# Balanced Fertilization Enhances the Nutritional Value and Flavor Profile of Tomato Fruits

**DOI:** 10.3390/foods13223599

**Published:** 2024-11-11

**Authors:** Wangxiong Li, Yang Zhang, Zhongqi Tang, Junwen Wang, Yue Wu, Jihua Yu

**Affiliations:** College of Horticulture, Gansu Agricultural University, Yingmen Village, Anning District, Lanzhou 730070, China; 18409319720@163.com (W.L.); 18409440280@163.com (Y.Z.); wangjw@st.gsau.edu.cn (J.W.); wuyue_gsau@163.com (Y.W.); yujihuagg@163.com (J.Y.)

**Keywords:** tomato, balanced fertilization, nutritional quality, polyphenols, volatiles

## Abstract

The tomato is a key fruit in China. However, the drive to produce higher-quality tomatoes has resulted in fertilizer overuse, soil degradation, and environmental pollution in recent years. Therefore, investigating the effects of balanced fertilization on the nutritional and flavor qualities of tomato plants is crucial. This study applied four fertilizer treatments to assess their effects on sugar and acid contents, sugar-metabolism-related enzyme activity, nitrate levels, ascorbic acid, pigments, polyphenols, and volatiles, and we performed a correlation analysis. The results showed that balanced fertilization increased glucose and fructose contents by 45% and 31% compared to CK (conventional fertilizer), while tartaric, citric, acetic, malic, and shikimic acid contents were reduced by 59%, 27%, 22%, 26%, and 4%, respectively. Additionally, balanced fertilization increased the activities of sucrose synthase (SS), sucrose phosphate synthase (SPS), acid invertase (AI), and neutral invertase (NI) by 58%, 26%, 19%, and 35%, respectively, compared to CK (conventional fertilizer) and upregulated the expression of *phosphoenolpyruvate carboxykinase* (*PEPCK*), *neutral invertase* (*NI*), *sucrose-phosphate synthase* (*SPS*), and *fructose-1,6-bisphosphatase* (*FBP*) genes. Moreover, balanced fertilization significantly enhanced the polyphenol content, as well as the diversity and concentration of volatiles. Correlation analysis confirmed that sugar-metabolism-related enzymes and genes were positively correlated with sugar fractions and negatively correlated with the organic acid content. Principal components analysis demonstrated that the balanced fertilization treatment was distinct from the other treatments, and all polyphenols, except for caffeic acid, were positively associated with balanced fertilization.

## 1. Introduction

The tomato (*Solanum Lycopersicon* L.), a globally important fruit crop, is increasingly produced in China, Japan, and Southeast Asia. As a key component of a healthy diet, it provides abundant sources of essential nutrients, including vitamin C, phenolic compounds (phenolic acids and flavonoids), minerals, and lycopene, which functions as both an antioxidant and a nutrient [1,2]. Tomatoes are often processed into ketchup or consumed raw [3]. Studies showed that appealing tomatoes are typically medium-sized with bright red or reddish-orange hues, and the color is regulated by pigments like lycopene [4]. Additionally, consumers prefer tomatoes that are flavorful, juicy, sweet, and have minimal seeds [5]. The fruit flavor is one of the most important factors for farmers and consumers, and the type and quantity of volatile substances are important in affecting the flavor quality of fruit. The various flavor substances of tomato fruit include alcohol, aldehyde, ester, ketone, hydrocarbon, and substances in another five categories, totaling more than 400 substances [6,7]. Natural tomatoes contain various essential compounds, including sugar–acid fractions and volatile flavor substances [8,9]. Among them, the synthesis and transformation of sugar–acid components are jointly regulated by SPS, SS, AI, NI enzymes, and genes related to sugar metabolism [10,11], which together influence the taste and flavor quality of fruits.

However, the reliance on fertilizer to boost yields has led to unscientific application rates, severely affecting the sustainable development of the fruit and vegetable industry [12]. In the rapid expansion of facility-based, off-season tomato cultivation in China, over-fertilization is prevalent. The methods and timing of fertilizer application are often unclear, and the application ratios of nitrogen, phosphorus, and potassium do not match crop requirements [13]. Excessive fertilizer use has increased input costs, wasted resources, reduced fertilizer efficiency, caused substrate erosion, and led to nutrient imbalances, ultimately reducing substrate fertility and increasing greenhouse gas emissions [14]. Therefore, regulating chemical fertilizer application, optimizing its use, and applying it based on crop nutrient uptake patterns are key strategies for efficiently utilizing resources, increasing yields, maintaining substrate fertility, and improving fruit and vegetable quality. Thus, the government has initiated policies to reduce chemical fertilizer use, to mitigate its negative impacts on the environment and human health while maintaining food security. New approaches, such as balanced fertilization, have gained public attention [15].

Balanced fertilizer application involves supplementing the appropriate proportions of nitrogen, phosphorus, potassium, and micronutrients based on the nutrient requirements of crop, the nutrient supply of soil before planting, and the fertilizer efficiency. The goal is to redistribute the nitrogen, phosphorus, and potassium levels based on current farm fertilizer usage, optimize their production and application, establish fertility observation points, and research a reasonable fertilizer distribution to reduce chemical fertilizer usage [16,17]. Reports indicate that balanced fertilizer application positively affects wheat yields, regardless of the fertilizer amount applied [17]. Additionally, balanced fertilization sustains soil microbial communities, enhances soil fertility and rice productivity in red paddy soils [18], improves tea yield and quality while reducing the environmental impact of cultivation [19], and decreases nitrogen losses with long-term use in maize–wheat systems [20].

However, few studies have focused on balanced nutrient fertilization in tomato cultivation under substrate conditions. Therefore, this experiment employed substrate cultivation, incorporating the existing amount of farmyard fertilizer and balanced fertilization to provide a practical basis for determining the optimal fertilizer application rates in tomato substrate cultivation. This approach has significant value for improving tomato quality during overwintering production in substrate-based systems.

## 2. Materials and Methods

### 2.1. Experimental Materials and Growth Conditions

The experiment was conducted using tomato Solanum Lycopersicon cv. 184. Seeds were germinated in black plug trays (12 × 6 hole) filled with seedling substrate (vermiculite: grass carbon at a 3:1 ratio), and then we planted tomato seedlings on 2 November and started the harvest on 19 February of the following year. Harvesting begins when the first cluster of fruits on the tomato plants turns pink, and the harvest is carried out once every five days until the end of the harvesting period.

The experiment employed substrate cultivation with a substrate volume ratio of cow manure: coconut coir: peat moss at 1:2:2. The nutrient contents of the substrate were as follows: total nitrogen (N) 4.15 g·kg^−1^, total phosphorus (P) 1.062 g·kg^−1^, total potassium (K) 1.68 g·kg^−1^, available nitrogen 52.81 mg·kg^−1^, available phosphorus 85.25 mg·kg^−1^, and available potassium 158.44 mg·kg^−1^. The pH value of the substrate was 7.4, and the electrical conductivity (EC) was 0.87 mS·cm^−1^. Additionally, after planting, a water and fertilizer integration machine was used for corresponding fertilizer treatments (Table 1) through drip irrigation, with 1.5 L of water per plant. Subsequently, the irrigation volume during plant growth was determined based on weather conditions and plant growth, typically once every 3–6 days.

### 2.2. Treatments

Four treatments were applied, and the fertilizer dosages are detailed in Table 1. Fertilization in this experiment followed local practices and dosages, based on the instructions for commercial fertilizers: (NH_4_)_2_HPO_4_ (46% nitrogen and 18% P_2_O_5_), 5Ca (NO_3_)_2_, NH_4_NO_3_·10H_2_O (15% nitrogen), and K_2_SO_4_ (52% K_2_O). Each treatment had five biological replicates, with 152 plants per replicate. The greenhouse temperature was maintained at 24–28 °C during the day and 7–16 °C at night, with humidity levels between 50 and 80%. The duration of sunlight was 10 h per day.

The first harvest began 162 days after transplanting, with subsequent harvests occurring every 5 days. Harvesting continued for two months, with three to four fruits collected at each developmental stage. Fruits were immediately frozen in liquid nitrogen and stored at −80 °C for further experiments.

### 2.3. Determination of Contents

#### 2.3.1. Measurement of Vitamin C and Nitrate Contents

Ascorbic acid or vitamin C (Vc) was determined using the sodium 2,6-dichlorophenol indophenol method, and nitrate content was determined using the salicylic acid method, along with the method referenced from Liu et al. [21].

#### 2.3.2. Determination of Sugar and Acid Contents

To determine sugar contents, a fresh sample (5 g) was taken and placed in a 30 mL centrifuge tube, and deionized water was added to 25 mL. Ultrasonic extraction was conducted for 60 min, then centrifuged at 10,000 rpm for 10 min, and the sample was filtered through a 0.22 μm water membrane, then finally added to the machine. The chromatographic conditions of sugar components were as follows: LC-NH2 column (460 mm × 250 mm); detector: differential refractive detector; mobile phase: acetonitrile–water = 75:25, and the mobile phase was degassed by ultrasonic vibration for 30 min; flow rate: 1 mL /min; column temperature: 30 °C; sample size: 20 μL.

For the determination of acid contents, a fresh sample of 5 g was taken and placed in a 30 mL centrifuge tube. Then, 25 mL of deionized water was added to it and centrifuged at 10,000 r/s for 10 min, then filtered through a 0.45 μm organic membrane and placed in the machine for measurement. Chromatographic conditions of acid components: Hi-Piex H (300 mm × 7.7 mm) column; detector: UV detector; detection wavelength: 210 nm; mobile phase: 10 mmol/L H_2_SO_4_; column temperature: 50 °C; flow rate: 0.4 ml·min^−1^; sample size: 20 μL [22].

#### 2.3.3. Determination of the Pigment Content of Tomato Fruits

0.5 g of dry sample was taken, and the carotenoids were extracted with 50 mL of petroleum ether and acetone (2:1, *v*/*v*) under ultrasonic conditions. The extract was collected into a brown bottle until all the colors were removed, and the combined filtrates were then transferred to a separatory funnel and washed twice with 250 mL of distilled water. After draining out the aqueous phase, it was removed via the addition of anhydrous sodium sulfate, and the filtrate was filtered under a vacuum with sand core funnel filtration, then poured into a round-bottomed flask. Petroleum ether extract was reduced to dryness by rotary evaporation at <45 °C. Finally, it was dissolved with 25 mL of acetonitrile: dichloromethane: methanol (55:20:25) and filtered through a 0.22 μm membrane on a machine. The HPLC18 column (250 mm × 4.6 mm, 5 μm, Waters Symmetry) was used at a column temperature of 25 °C. The flow rate was 1.2 mL·min^−1^, and the mobile phase was acetonitrile: dichloromethane: methanol 55:20:25. We used a Waters liquid chromatography system equipped with a 1525 pump and a 2998 photodiode array detector for the HPLC analysis. A 10 μL sample was aspirated, and the pigment content was detected at 450 nm [23].

#### 2.3.4. Determination of Polyphenols’ Composition

In this step, 0.1 g of dry powder was weighed, 2 mL of methanol was added, and the supernatant was centrifuged at 8000 rpm for 10 min at 4 °C. The supernatant was filtered through a 0.22 μm filter membrane and then detected by HPLC. HPLC conditions: A Waters liquid chromatography system equipped with a 1525 pump and a 2998 photodiode array detector was used (250 mm × 4.6 mm, 5 μm, Waters Symmetry); column temperature: 30 °C; flow rate: 1.1 mL/min; mobile phase: methanol (A) and 1% acetic acid (B). The analytes were detected at 240 nm for p-hydroxybenzoic acid, protocatechuic acid, quercetin, chlorogenic acid, and rutin; 280 nm for cinnamic acid, 4-coumaric acid, gallic acid, naringenin, benzoic acid, and ferulic acid; and 322 nm for erucic acid, caffeic acid, artichoke, and 2,5-dihydroxybenzoic acid [24].

#### 2.3.5. Determination of Enzyme Activity Associated with Sugar Synthesis

Acid invertase (AI), neutral invertase (NI), sucrose phosphate synthase (SPS), and sucrose synthase (SS) were measured by a reagent kit (Beijing Solarbio Science &Technology Co., Ltd., Beijng, China). The experimental method was according to the manufacturer’s instructions.

#### 2.3.6. Determination of Volatile Components

The volatile components were analyzed by gas chromatography–mass spectrometry (GC-MS), referring to the method of Wei et al. [25] with slight modifications. Tomato fruits were washed with ultrapure water after removing the seeds and dried, and then they were quickly beaten into a homogenate with a homogenizer. Next, 9 g of homogenate was accurately weighed and placed in a 20 mL headspace injection vial, to which 1.5 g of anhydrous sodium sulfate, 10 µL of 88.2 mg·L^−1^ 2-octanol standard (chromatographic purity), and a magnetic stirring rotor were added, and the cap was quickly screwed on. The sample was stirred in a magnetic stirrer at a constant temperature of 50 °C for 10 min, and then the extraction needle was inserted into the injection bottle for headspace solid-phase extraction at 50 °C for 30 min. After extraction, the extraction needle was inserted into the chromatographic gasification chamber, and the sample was resolved for 3 min and then finally analyzed by GC-MS. Chromatographic conditions: DB-WAX flexible quartz capillary column (20 m × 0.18 mm, 0.18 μm); inlet temperature of 230 °C; carrier gas: helium with purity ≥99.999% at a flow rate of 1.0 mL·min^−1^ and a shunt ratio of 30/1; injection mode: no shunt injection, and the shunt valve was opened after 1 min; programmed temperature increase: the initial temperature of 40 °C was increased to 140 °C at a rate of 3.5 °C·min^−1^ and then at a rate of 30 °C·min^−1^ to 190 °C, where it was maintained for 3 min; ion source temperature 200 °C; transmission line temperature 190 °C; ionization mode: EI; electron energy: 70 eV; scanning mode: full scan; scanning mass range: 35~500 u.

After the volatile substances in tomato fruit were analyzed and identified by GC-MS, each chromatographic peak was searched with a computer and compared with the standard mass spectrometry library (NIST 2014), supplemented with manual qualitative identification of the corresponding compounds. Only volatile substances with positive and negative matches greater than 800 were identified, with reference to the matches of the mass spectra as well as volatile substance compositions reported in the relevant literature. The volatiles extracted from tomato fruits were quantitatively analyzed using standards. The content of each volatile component was calculated according to the following formula: volatile content (µg/kg) = (sample peak area/internal standard peak area × internal standard mass/(µg) × 1000)/(sample mass/(kg)).

#### 2.3.7. qRT-PCR Analysis

The sample was ground to fine powder using liquid nitrogen. Total RNA was extracted from tomato using a Trizol kit (Tiangen Biochemical Technology Co., Ltd. Shanghai, China) according to the manufacturer’s instructions. The concentration and purity of RNA were detected by a spectrophotometer, respectively. The cDNA was synthesized by a Prime Script TM RT reverse transcription kit. According to the full-length sequence of related enzyme genes, the target gene-specific primers were designed using Beacon Designer 7.9 software (Table 2). Quantitative real-time PCR (qRT-PCR) was conducted using the Power SYBR^®^ Green PCR Master Mix (Tiangen Biochemical Technology Co., Ltd. Shanghai, China) according to the manufacturer’s instructions with gene-specific primers and 100 ng of cDNA, and the total reaction system was 20 μL. Three parallel tube replicates were set up in each group. The actin gene suitable for quantitative fluorescence PCR in tomato was used as the internal reference gene (Solyc03g078400.2). The amplification program consisted of initial denaturation at 95 °C for 10 min, followed by 40 subsequent cycles of denaturation at 95 °C for 15 s, and annealing and extension at 60 °C for 1 min. QRT-PCR was carried out on a Step One plus™ Thermal Cycler and analyzed using Step One v2.3 software (Applied Biosystems, Foster City, CA, USA), and the data were analyzed by the 2^ΔΔCt^ method.

### 2.4. Statistical Analysis

Data were analyzed using one-way analysis of variance (ANOVA) in SPSS software (version 22.0; SPSS Institute Inc., Chicago, IL, USA); significant differences were compared using Duncan’s multiple range test (*p* < 0.05). Correlation analysis and principal component analysis (PCA) were performed using Origin 2021 (Origin Inc., San Francisco, CA, USA). Results are presented as the mean ± standard error (SE). In all analyses, a probability value below 0.05 was considered statistically significant (*p* < 0.05).

## 3. Results

### 3.1. Balanced Fertilization Altered the Nitrate and Ascorbic Acid Levels

Fertilization is the primary factor affecting nitrate content. Notably, nitrogen application significantly affects nitrate content. In this experiment (Figure 1A), the balanced fertilization treatment (T1 treatment) significantly increased nitrogen content compared to the control (CK). To validate the predictions of balanced fertilization, a lower treatment level (T2) was implemented. The nitrate content in the balanced fertilization treatment (T1) was 165.19 mg·kg^−1^, while in the 10% reduce treatment (T2), it was 161.15 mg·kg^−1^. Compared to the control (CK) value of 156.40 mg·kg^−1^, these values represent increases of 5.3% and 2.9%, respectively. This indicates that nitrogen regulation under balanced fertilization does not result in significant nitrate accumulation in fruits.

The effects of nitrogen and potassium on vitamin C (Vc) were also substantial. The Vc contents in treatments T1 and T2 were 11.71 mg·kg^−1^ and 9.33 mg·kg^−1^, respectively, which were higher than CK (7.87 mg·kg^−1^, Figure 1B) by 32.82% and 15.65%. However, high nitrogen application (T1) increased Vc content while also leading to a minor accumulate of nitrate (Figure 1A,B). The nitrate levels in T1 were significantly higher than in CK0, CK, and T2 by 52.86%, 32.82%, and 20.41%, respectively, while the nitrate and ascorbic acid contents in T2 were 32.27% and 42.87% higher than in CK0. This study demonstrates that a judicious fertilizer combination and scientifically balanced application significantly enhance Vc accumulation in fruits.

### 3.2. Balanced Fertilization Changed the Lycopene, Lutein, and Violaxanthin

Nitrogen, phosphorus, and potassium fertilizers directly or indirectly regulate tomato fruit coloration by affecting pigments’ composition. Figure 2 demonstrates the impact of balanced fertilization on lycopene, lutein, and violaxanthin contents. The pigments in fruits under fertilization were significantly higher than in the non-fertilization control (CK0). Specifically, lycopene contents in the T1 treatment were 58.02%, 38.87%, and 16.37% higher than in CK0, CK, and T2, respectively (Figure 2A). Lutein and violaxanthin levels in T1 were 294.90 mg·100 g^−1^ and 4.92 mg·100 g^−1^, representing increases of 15.22% and 27.06% compared to CK (Figure 2B,C). However, the lutein content under the T2 treatment was 7.32% lower than CK. This indicates that lycopene, lutein, and violaxanthin levels increased with balanced fertilization, but all three pigments decreased under a 10% reduction in fertilization.

### 3.3. Balanced Fertilization Influenced the Sugar and Acid Components

The balance of sugar and acid in tomato fruit determines its flavor. Figure 3A,B show the effects of balanced fertilization on the primary acids (acetic, citric, malic, tartaric, oxalic, and shikimic) and main sugars (glucose, fructose, sucrose) in tomatoes.

T1 treatment significantly increased fructose and glucose contents, while reducing tartaric, citric, acetic, malic, and shikimic acids. Under balanced fertilization or its reduction (T1, T2), the six acids decreased, with T1 showing significant differences compared to T2. Glucose, fructose, and citric acid were the most abundant components. Glucose contents in the T1 and T2 treatments were 31.24 mg·100 g^−1^ and 25.93 mg·100 g^−1^, respectively, 45.14% and 17.01% higher than CK (21.52 mg·100 g^−1^). The fructose content in T1 was 23.71 mg·100 g^−1^, 11.57% higher than CK (21.24 mg·100 g^−1^, Figure 3A). Citric acid contents in the T1 and T2 treatments were 5.24 mg·100 g^−1^ and 6.22 mg·100 g^−1^, respectively, representing reductions of 37.60% and 15.92% compared to CK (7.21 mg·100 g^−1^). Compared to CK, the tartaric, acetic, and malic acid contents in T1 decreased by 59%, 22%, and 26%, respectively, but with no significant changes in shikimic and oxalic acids (Figure 3B). These results demonstrate that balanced fertilization increased the sugar content and reduced the acid content.

### 3.4. Balanced Fertilization Influenced Enzymatic Activities Related to Sucrose Metabolism

SS, SPS, NI, and AI are key enzymes in plant sugar metabolism, jointly regulating the synthesis, breakdown, and transport of sucrose. Enzymatic activities under these four different fertilizer treatments are shown in Figure 4. All four enzymes followed a similar trend, with higher activity in the balanced fertilization treatment (T1) compared to treatments with SS, SPS, NI, and AI, with statistically significant differences except for SPS. Under balanced fertilization (T1), SS, SPS, AI, and NI activities were 58%, 26%, 19%, and 35% higher, respectively, compared to CK. AI and NI activities under T2 were also higher than in CK. However, SS activity in T2 was lower than in CK. These results suggest that balanced fertilization promotes the activities of sucrose synthase, sucrose phosphate synthase, and invertase.

### 3.5. Balanced Fertilization Influenced Gene Expression Related to Sucrose Metabolism

The genes for neutral invertase (*NI*), pyruvate kinase 1 (*PK1*), pyruvate kinase 2 (*PK2*), hexokinase 1 (*HXK1*), phosphoenolpyruvate carboxykinase (*PEPCK*), acid invertase (*AI*), hexokinase 2 (*HXK2*), fructose-1,6-bisphosphatase (*FBP*), sucrose-phosphate synthase (*SPS*), and sucrose synthase (*SS*) encode enzymes involved in sugar metabolism pathways, including sucrose conversion, gluconeogenesis, and glycolysis. The results presented in Figure 5 indicate that compared to CK0, CK treatment significantly up-regulated the genes NI, AI, PK2, HXK1, PEPCK, FBP, SPS, and SS, while down-regulating PK1. Compared to CK treatment, the T1 treatment significantly upregulated the genes *NI*, *AI*, *PK1*, *HXK1*, *PEPCK*, *FBP*, *SPS*, and *SS* by 53.62%, 125.34%, 471.43%, 89.33%, 80.61%, 508.03%, 120.28%, and 220.88%, respectively. The T2 treatment significantly downregulated the genes *AI*, *PK1*, *HXK1*, *FEP*, and *SPS* compared to the T1 treatment. Thus, balanced fertilization activates genes that encode enzymes involved in sugar metabolism and enhances overall sugar metabolism.

### 3.6. Pearson’s Correlation Analysis and Principal Component Analysis of Nutritional Quality in Tomato Fruits

Pearson correlation analysis is widely employed to assess the degree of correlation between two parameters. As illustrated in Figure 6A, several sets of significant positive correlations (*p* < 0.05) exist among nutritional quality parameters. The figure’s color indicates that six acids (acetic, citric, malic acid, tartaric, oxalic, and shikimic) and gene *PK1* are negatively correlated with other parameters, whereas the remaining parameters exhibit positive correlations with each other. Several parameters exhibited significant positive correlations (*p* < 0.05), including nitrate, which was highly positively correlated with the SPS gene (*r* = 1.00); ascorbic acid, which was highly positively correlated with glucose (*r* = 1.00); violaxanthin, which was highly positively correlated with AI and NI (*r* = 1.00); AI, which was highly positively correlated with NI (*r* = 1.00); and the NI gene, which was highly positively correlated with the PEPCK gene (*r* = 1.00). Ascorbic acid was also significantly positively correlated with lycopene and NI (*r* = 0.99); lycopene and AI were significantly positively correlated (*r* = 0.99); violaxanthin was significantly positively correlated with glucose (*r* = 0.99); and glucose exhibited a significant positive correlation with AI and NI (*r* = 0.99).

Figure 6B presents the PCA classification model utilized to analyze the effects of different treatments on the nutritional quality of tomato fruits. The first and second principal components (PCs) captured most of the variation observed across treatments. These two components accounted for 88.8% of the total variance, with PC1 explaining 67.7% and PC2 explaining 21.1%. The loading diagram (Figure 6B) indicates that ascorbic acid, violaxanthin, lutein, lycopene, glucose, nitrate, AI, NI, SS, SPS, PEPCK, NI, SS, and HXK1 genes are key factors influencing PC1, while shikimic and tartaric acids are key factors influencing PC2. When examining the projections of each indicator on PC1 and PC2, we observed that the *PEPCK*, *NI*, *SS*, and *HXK1* genes are closer to T1, while ascorbic acid, violaxanthin, lutein, lycopene, glucose, nitrate, and *AI*, *NI*, *SS*, and *SPS* genes are closer to T2. This suggests a strong correlation between the indicators and treatments T1 and T2. Furthermore, the acute angles between these indicators indicate a positive correlation among them. Based on PC1, the four treatments were delivered to three groups: CK0 and CK, and T1 and T2. This classification highlighted the significant differences in nutritional quality indicators between T1 and T2, as well as between CK0 and CK.

### 3.7. Balanced Fertilization Influenced Polyphenol Composition

Polyphenols serve as important antioxidants in tomato fruits. Table 3 displays the detection of three flavonoids (quercetin, rutin, and naringenin) and ten phenolic compounds (p-hydroxybenzoic acid, protocatechuic acid, cinnamic acid, 4-coumaric acid, gallic acid, benzoic acid, ferulic acid, caffeic acid, cynarin, and 2,5-dihydroxybenzoic acid) in this experiment. The results indicated that the CK treatment significantly increased the levels of quercetin, protocatechuic acid, cinnamic acid, 4-coumaric acid, and gallic acid compared to the CK0 treatment. Compared to the CK treatment, the T1 treatment significantly increased the contents of quercetin, rutin, naringenin, p-hydroxybenzoic acid, protocatechuic acid, 4-coumaric acid, gallic acid, benzoic acid, ferulic acid, caffeic acid, cynarin, and 2,5-dihydroxybenzoic acid by 67.06%, 35.25%, 57.76%, 77.58%, 130.88%, 53.85%, 53.96%, 10.63%, 328.79%, 64.59%, and 260.29%, respectively. However, the caffeic acid content significantly decreased by 14.81%. In contrast, the T2 treatment resulted in a significant reduction in the contents of rutin, p-hydroxybenzoic acid, protocatechuic acid, ferulic acid, caffeic acid, cynarin, and 2,5-dihydroxybenzoic acid compared to the T1 treatment.

### 3.8. Pearson’s Correlation Analysis and Principal Component Analysis of Polyphenols in Tomato Fruits

Pearson correlation analysis is frequently employed to assess the strength of correlation between two parameters. Figure 7A illustrates significant positive correlations (*p* < 0.05) between flavonoids and phenolic acids. For instance, naringenin and p-hydroxybenzoic acid exhibited a strong positive correlation (*r* = 0.99), while quercetin was positively correlated with protocatechuic acid (*r* = 0.98). Cynarin displayed significant positive correlations with both 2,5-dihydroxybenzoic acid and naringenin (*r* = 0.97). Additionally, cynarin was positively correlated with ferulic acid, p-hydroxybenzoic acid, and protocatechuic acid (*r* = 0.96). Moreover, p-hydroxybenzoic acid demonstrated a positive correlation with protocatechuic acid (*r* = 0.96), and naringenin was positively correlated with ferulic acid and 2,5-dihydroxybenzoic acid (*r* = 0.95).

Figure 7B presents the PCA classification model utilized to analyze the effects of different treatments on polyphenols in tomato fruits. The first and second principal components (PCs) captured the observed variation across treatments, accounting for 89.6% of the total variance, with PC1 explaining 70.5% and PC2 19.1%. Notably, all polyphenol fractions, except caffeic acid, clustered in the positive range of PC1. This outcome indicates that these eleven metrics exhibit linear and positive correlations, thereby confirming the findings depicted in Figure 7A. The loading diagram revealed that cynarin, ferulic acid, 2,5-dihydroxybenzoic acid, and naringenin significantly influenced the first principal component, while caffeic acid was a key factor affecting the second principal component. Consequently, these compounds are regarded as representative factors reflecting the polyphenolic composition of tomato fruits under varying fertilizer treatments. In terms of treatment separation, CK0, CK, T1, and T2 treatments were distinguished along PC1. T1 and T2 were distinctly separated from CK0 and CK along PC1. Compared to caffeic acid, the other indices demonstrated a closer proximity to T1, indicating a strong correlation with the T1 treatment.

### 3.9. Balanced Fertilization Influenced Volatile Components

The types and quantities of aromatic substances present in tomatoes contribute to their unique flavor and aroma. As shown in Table 4, a total of 47 volatile compounds were detected using headspace solid-phase microextraction–gas chromatography–mass spectrometry (HS-SPME/GC-MS) across the four treatments, including 15 alcohols, 13 aldehydes, 5 esters, 7 ketones, 2 hydrocarbons, 3 phenols, and 2 other compounds.

In terms of total volatile compounds, the T1 treatment exhibited the highest concentration at 2644.02 µg kg^−1^, followed by the T2 treatment at 2047.53 µg kg^−1^. The CK and CK0 treatments had the lowest volatile compound concentrations. The T1 treatment significantly increased the contents of most alcohols, particularly ethanol, 1-hexanol, and 3-hexen-1-ol (Z)-, with T1 showing increases of 57.68%, 134.28%, and 315.59% compared to CK. There was no significant difference between CK0 and CK. Regarding aldehydes, T1 treatment elevated the levels of hexanal (n-), pent-(2E)-enal, cyclohexane, 1,1’-(2-methyl-1,3-propanediyl) bis-, and 2-cyclohexen-1-ol, with significant increases of 78.51%, 116.00%, and 75.90%, respectively, compared to CK. Among the esters, hex-(3Z)-enyl acetate and butyl salicylate saw significant increases of 220.29% and 520.29%, respectively, in the T1 treatment relative to CK. Furthermore, T1 significantly enhanced the levels of penten-3-one, cycloheptadecanone, hept-5-en-2-one (6-methyl-), and trans-α-ionone compared to CK. Additionally, T1 treatment significantly increased the contents of phenol, 2-methylethyl acetate, and butyl salicylate compared to CK.

As presented in Table 5, a total of 11 characteristic aroma compounds were detected in the T1 treatment, with 10 of these exhibiting significantly higher concentrations than those found in the CK0, CK, and T2 treatments. Notably, (E)-2-heptenal was not detected in the CK0, CK, and T2 treatments but was present in the T1 treatment at a concentration of 2.23 µg·kg^−1^. Additionally, 1-penten-3-ol, penten-3-one, hept-5-en-2-one (6-methyl-), trans-α-ionone, and 2-isobutylthiazole contributed fruity flavors to the tomato fruit. Trans-α-ionone and 2-isobutylthiazole also provided aromatic and green notes. In contrast, l-phenylethyl alcohol imparted a floral flavor, while (Z)-3-hexen-1-ol, 3-hexenal, hexanal (n-), (E)-2-hexenal, and (E)-2-heptenal contributed grassy and green flavors.

## 4. Discussion

Fertilization promotes plant growth; however, it also disrupts soil structure. Studies have shown that irrational fertilization patterns and durations decrease soil organic carbon content [26], whereas balanced fertilization enhances soil fertility and increases rice productivity [18]. Therefore, adopting a rational fertilization pattern is essential.

This study examined the significant differences in the effects of balanced versus traditional fertilization methods on the primary and secondary metabolites of tomato fruits cultivated in a controlled environment. Our findings indicated that compared to CK, the T1 treatment significantly increased the soluble sugar and ascorbic acid contents while reducing organic acid levels in tomato fruits (Figure 1 and Figure 3). Malundo et al. clearly demonstrated that the sweetness and sourness of tomatoes are significantly influenced by their sugar and acid levels [27]. Furthermore, studies on ‘Huang-guan’ pears cultivated in desert regions highlighted that balanced fertilization (medium levels of nitrogen (N), phosphorus pentoxide (P_2_O_5_), and sulfur (S)) enhances ascorbic acid (ASA) and vitamin C (Vc) while maintaining productivity [28], which was consistent with our previous study finding that balanced fertilization significantly increased the contents of iron, magnesium, and calcium elements in tomato fruits [29]. Additionally, research has indeed shown that balanced fertilization plays a positive role in modulating sugar production and enhancing the sweetness of tomatoes by improving the soil structure [30]. Similarly, our findings suggest that balanced fertilization modifies the soil structure, thereby enhancing nutrient absorption in tomatoes and increasing the sugar content while reducing acid levels in the fruit; meanwhile, the increased minerals (N, Fe, Ca, Mg) improve the metabolism of tomato plants and their photosynthesis, thereby increasing the absorption of nutrition for plants [31].

Furthermore, sugars were synthesized through photosynthesis in the leaves and subsequently accumulated in the fruits through various physiological processes [32]. Once inside the fruit, sucrose is converted into glucose and fructose by neutral invertase (NI) enzymes, which can be further phosphorylated into the glycolytic pathway [33]. Sucrose synthase (SS) enzymes catalyze the reversible conversion of sucrose into fructose, glucose, and UDP-glucose (UDPG), making them crucial for sucrose synthesis. Additionally, sucrose phosphate synthase (SPS) enzymes are essential for the irreversible conversion of UDP-glucose (UDPG) and fructose-6-phosphate into sucrose. [34,35]. Cytosolic pyruvate kinase (PK) serves as a crucial glycolytic enzyme in the tricarboxylic acid (TCA) cycle and regulates carbohydrate flux in Arabidopsis thaliana [36]. Furthermore, the genes encoding AI, NI, SS, and SPS are consistent with the results of correlation and principal component analyses, indicating that sugar conversion-related genes are positively correlated with enzyme activity (Figure 6). Reports have shown that the NI gene decomposes sucrose to maintain a consistent sucrose concentration between sources in tomatoes [37]. Phosphoenolpyruvate carboxykinase (*PEPCK*) plays a positive role in determining the sugar-to-acid ratio in ripening fruit [38]. The gene *PbHXK1* negatively correlated with sugar content during the development of pear fruit [39]. Sucrose phosphate synthase (*SPS*) is involved in the regulation of carbon partitioning in leaves [40]. Overexpression of fructose-1,6-bisphosphatase (*FBP*) enhances source capacity, leading to increased growth in transgenic plants [41]. Additionally, suppressing the gene *SIHXK1* affects leaf senescence and plant growth and development in tomatoes by influencing starch turnover [42]. Figure 4 and Figure 5 illustrate that the T1 treatment significantly increased the activity of sucrose-converting enzymes and upregulated genes associated with photosynthesis. These findings suggest that balanced fertilization enhances fertility by modifying the soil microbial structure, thereby improving the photosynthetic capacity of tomato fruits, and promoting sugar accumulation.

Additionally, the nitrate content serves as a crucial indicator of fruit and vegetable health and safety. Research indicates that elevated nitrate levels primarily result from excessive nitrogen in fertilizers. When nitrogen fertilizer application is excessive, plants absorb an excessive amount of nitrogen, leading to its accumulation in plant tissues [43]. In this experiment, the nitrate content ranged from 137.75 to 165.16 mg·kg^−1^, with no significant difference observed between balanced and conventional fertilization treatments (Figure 1). Furthermore, lycopene, lutein, and violaxanthin are carotenoids that serve as the primary pigments responsible for the typical yellow, orange, and red coloration in fruits. These pigments also play crucial roles in plant growth, development, and responses to environmental stimuli [44]. In green photosynthetic tissues, carotenoids engage in photosystem assembly, light trapping, and photoprotection. Additionally, carotenoids are essential nutrients in the human diet, serving as the primary source of vitamin A and acting as natural antioxidants [45]. Chicken feather compost delivered high nutrient levels, including organic carbon (1.83%), nitrogen (7.33%), potassium (4.40%), sulfur (19.69 ppm), zinc (4.96 ppm), boron (0.59 ppm), and iron (6.62 ppm). This compost also enhanced the lycopene and β-carotene contents in tomato fruits [46]. Under T1 treatment, the levels of lycopene, lutein, and violaxanthin significantly increased (Figure 2). This enhancement was attributed to the balanced fertilization strategy, which provided tomatoes with suitable nutrients that promoted the conversion of carotenoid compounds during the fruit color transition, thereby accelerating fruit coloration.

Polyphenols, which include flavonoids and phenolic acids, are a class of natural antioxidants that play a vital role in protecting human health [47]. Polyphenols are derived from carotenoids, which are converted through the shikimate pathway. These phytochemicals are widely distributed in plants and serve various biological functions, including responses to biotic and abiotic stressors. Balanced fertilization, providing adequate supplies of nitrogen (N), phosphorus (P), and potassium (K), increased polyphenol levels in artichokes, particularly 1,5-di- and 5-O-caffeoylquinic acid, as well as apigenin 7-O-glucuronide. However, this effect was influenced by genetic factors and soil type [48]. Fertilizers, particularly nitrogen (N) fertilizers, significantly influence polyphenol content. Typically, higher polyphenol levels are observed when lower amounts of N fertilizer are applied to the soil [49]. These findings align with our observations that balanced fertilization reduced nitrogen fertilizer content compared to the control group (CK), thereby promoting polyphenol accumulation and enhancing the antioxidant levels in the fruit (Table 3). This finding is further supported by the correlation analysis and principal component analysis, which revealed a positive correlation among all indicators except caffeic acid. Moreover, the T1 treatment was positioned similarly to the principal components (Figure 7).

Aroma is a crucial organoleptic quality of tomato fruits, produced by the collaboration of over 400 volatile aromatic substances. It significantly influences commodity grading and consumer acceptance. The types and concentrations of these substances vary, contributing to the distinct flavors of different tomato varieties. The forty volatile substances primarily include alcohols, aldehydes, ketones, esters, terpenoids, and sulfur-containing compounds. Sixteen of these substances have log-threshold units greater than 0, identifying them as significant contributors to the aroma of tomato fruits [50]. However, several factors influence volatile compounds, including plant species, cultivars, developmental stages, and root colonization by soil microorganisms [51]. Field-grown ‘V. R. Moscow’ tomatoes have been found to contain higher levels of aromatics, including 3-methylbutanol, 2-methyl-3-hexanol, 3-methylbutyraldehyde, saccharaldehyde, benzaldehyde, isoamyl acetate, methyl salicylate, and α-pinene, compared to greenhouse-grown tomatoes. Additionally, temperature and light conditions during different cultivation seasons affected aromatic synthesis: 48 kinds of volatile compounds were detected in spring crops, while only 37 were identified in fall/winter crops [52]. In this study, a total of 48 aromatic substances were detected. Among them, 40 were identified in the CK0 treatment, which did not include (E)-2-Heptenal, 1-Heptanol, (Z)-4-Decenal, 1-Butanol, 3-methyl acetate, and Methyl N-phenyl carbamate. In the T1 treatment (balanced fertilization), 45 substances were detected, while p-Xylene and tridecane were unique to the T2 treatment (10% reduction in balanced fertilization) (Table 3). Therefore, we hypothesize that the significant increase in volatiles resulting from balanced fertilization may be attributed, in part, to the cultivation of tomatoes in winter and spring crops in greenhouses. Additionally, it may be related to changes in the root colonization structure of soil microorganisms due to balanced fertilization.

Furthermore, characteristic aroma compounds emit unique scents. C6 aroma compounds impart a grassy and green flavor to tomato fruits [53]. C5 aromatics (such as pentanol, 1-penten-3-ol, pentanal, trans-2-pentenal, and 1-penten-3-one) contribute fruity flavors, and their levels correlate strongly with consumer preferences for tomatoes. Additionally, 2-phenylethanol and 2-phenylethylaldehyde provide floral flavors, while 1-nitro-2-ethylbenzene contributes an earthy flavor. Compounds such as 2-methylbutanol, 3-methylbutanol, 2-methylbutanal, and 3-methylbutyraldehyde impart caramel flavors, while isobutyl acetate and isovaleronitrile provide spicy and fruity notes, respectively [11]. Notably, 2-isobutylthiazole, a unique aromatic compound in tomatoes, contributes a fruity and green flavor [54]. In this study, 11 characteristic aromatic substances were identified, with significantly higher levels of volatile compounds observed under the T1 treatment compared to other treatments, except for 1-Penten-3-ol. The dominant flavors were the grassy and green notes of C6 aromatic substances, followed by fruity flavors from ester and thiazole aromatic compounds (Table 4).

## 5. Conclusions

In summary, the balanced fertilization treatment (T1) (N 38.38 kg·667 m^−2^, P_2_O_5_ 16.45 kg 667 m^−2^, K_2_O 53.18 kg·667 m^−2^) significantly enhanced sugar metabolism in tomato fruits, which contributed to the upregulation of key genes involved in sugar metabolism, significantly increased vitamin C (Vc) content, and reduced nitrate levels. Furthermore, the T1 treatment significantly enhanced the accumulation of lycopene, lutein, and violaxanthin, balanced the ratio of sugar and acids, and increased polyphenolic flavonoids and characteristic volatiles, improving the taste and nutritional quality of the fruits. This fertilization mode is recommended for overwintering tomatoes grown in facility substrates, effectively maximizing the benefits and optimizing fertilizer use.

## Figures and Tables

**Figure 1 foods-13-03599-f001:**
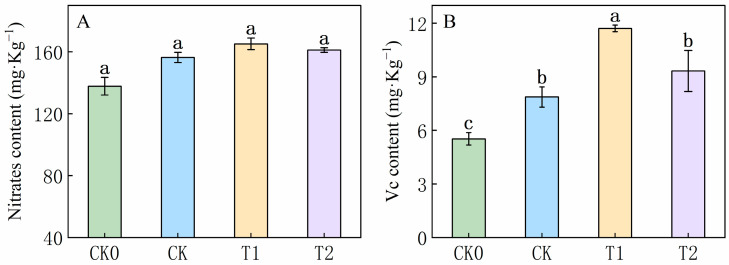
Effects of balanced fertilization on nitrate content (**A**) and ascorbic acid content (**B**) in tomato fruits. CK0 indicates no fertilizer; CK denotes conventional fertilizer (based on local practices), T1 indicates balanced fertilizer, and T2 refers to 10% less than the balanced fertilizer (10% less than T1). Values represent the mean of five biological replicates; different letters indicate significant differences (Tukey’s test, *p* < 0.05).

**Figure 2 foods-13-03599-f002:**
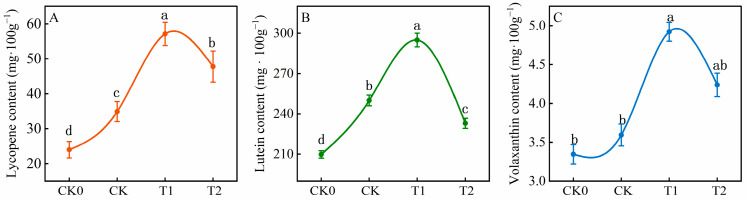
Effect of balanced fertilization on lycopene, lutein, and violaxanthin contents (**A**–**C**) in tomato fruit. CK0 represents no fertilizer, CK represents conventional fertilization (local fertilization practices), T1 represents balanced fertilization, and T2 represents 10% less than the balanced fertilization (T1). Values represent the mean of five biological replicates. Different letters indicate significant differences (Tukey’s test, *p* < 0.05).

**Figure 3 foods-13-03599-f003:**
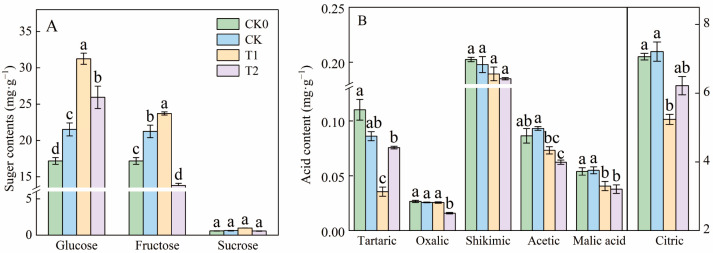
Sugar and acid contents under different fertilizer treatments (**A**,**B**). CK0 represents no fertilizer, CK represents conventional fertilization (local fertilization practices), T1 represents balanced fertilization, and T2 represents 10% less than balanced fertilization (T1). Values are the mean of five biological replicates. Different letters indicate significant differences (Tukey’s test, *p* < 0.05).

**Figure 4 foods-13-03599-f004:**
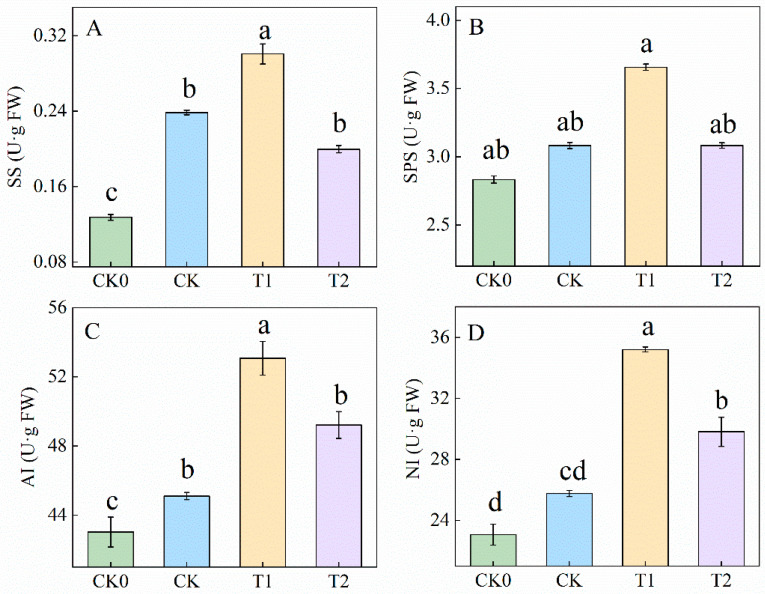
Enzyme activities related to sugar metabolism under different fertilizer treatments. Panels (**A**–**D**) represent the activities of SS, SPS, AI, and NI enzymes. CK0 represents no fertilizer, CK represents conventional fertilization (local fertilization practices), T1 represents balanced fertilization, and T2 represents 10% less than balanced fertilization (T1). Values are the mean of five biological replicates. Different letters indicate significant differences (Tukey’s test, *p* < 0.05).

**Figure 5 foods-13-03599-f005:**
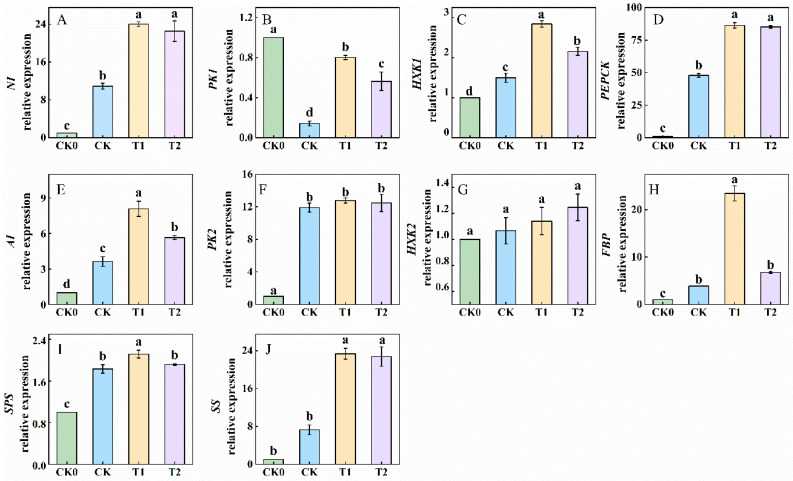
Expression of ten genes under different fertilizer treatments. Panels (**A**–**J**) represents the genes neutral invertase (*NI*), pyruvatekinase1 (*PK1*), pyrivatelkinase2 (*PK2*), hexokinase1 (*HXK1*), phosphoenolpyruvate carboxykinase (*PEPCK*), acid invertase (*AI*), hexokinase2 (*HXK2*), fructose-1,6-bisphosphatase (*FBP*), sucrose-phosphate synthase (*SPS*), and sucrose synthase (SS), respectively. CK0 represents no fertilizer, CK represents conventional fertilization (local fertilization practices), T1 represents balanced fertilization, and T2 represents 10% less than balanced fertilization (T1). Values are the means of five biological replicates. Different letters indicate significant differences (Tukey’s test, *p* < 0.05).

**Figure 6 foods-13-03599-f006:**
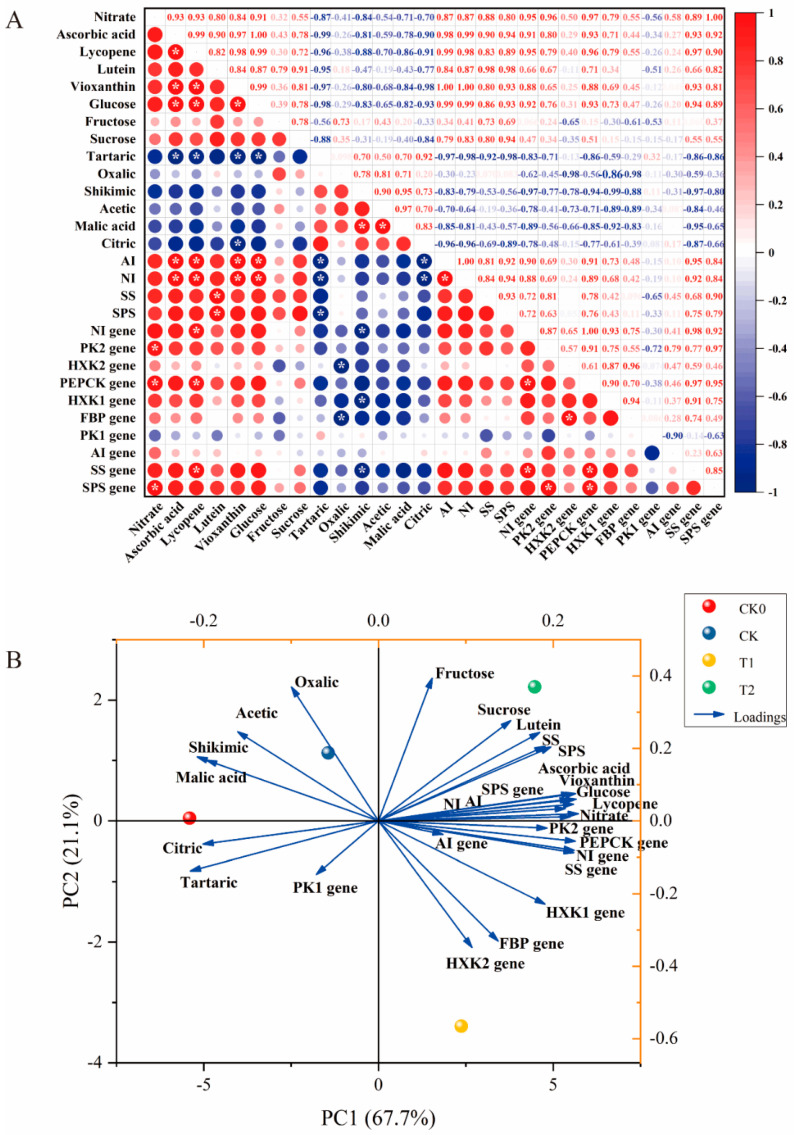
A Pearson correlation analysis (**A**) and principal component analysis (**B**) were conducted to assess the nutritional quality of tomato fruits.

**Figure 7 foods-13-03599-f007:**
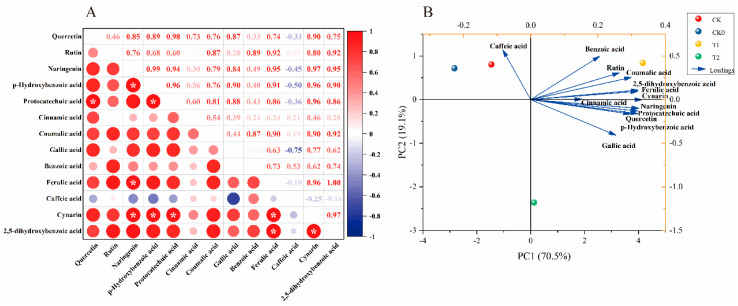
Pearson’s correlation analysis (**A**) and principal component analysis (**B**) were conducted to evaluate the polyphenol composition in tomato fruits.

**Table 1 foods-13-03599-t001:** Experimental setup for design and fertilizer dosages.

Treatments	Fertilizer Dosage (kg/ha)		
	N	P_2_O_5_	K_2_O
CK0	0	0	0
CK	355.5	576	1093.5
T1	575.7	246.75	797.7
T2	503.7	216	697.5

Note: CK0 represents no fertilizer, CK represents conventional fertilizer (based on research of local fertilization), T1 represents balanced fertilizer, T2 represents 10% less than balanced fertilizer (10% less than T1).

**Table 2 foods-13-03599-t002:** The primer of 11 genes.

Gene Number	Name	Forward Primer (5′-3′)	Reverse Primer (5′-3′)
Solyc03g121070.2	*HK1*	GCACTATACAGAATACAGGATG	AATGAGAGGCAGCAAGAA
Solyc06g066440.2	*HK2*	ATCCAATACCTCCTTGAAGA	ATACAATCCGCCATCCAT
Solyc01g106010.2	*FBP*	CTCTTGACACATCCTAACATC	ATTGCTACGCCACCATAT
Solyc01g049650.2	*PK1*	CCTGCTGAGTCTACGAAT	ATAATCTTAACCACCGATGC
Solyc01g106780.2	*PK2*	TGGTCAGGTGGAAGTTAT	CCAGAATCTCGGAAGGTA
Solyc12g088160.1	*PEPCK*	TGGTCAGGTGGAAGTTAT	CCAGAATCTCGGAAGGTA
Solyc07g045110.1	*SPS*	TATTCGTCCTTCCATTCTGA	GTCTTCATCCTCAACAACAA
Solyc02g081300.2	*SS*	GCTCAAGGACAGGACTAA	GCTCATACATCTTCTTCATCTC
Solyc03g083910.2	*AI*	CGGAATTGGATTGTGGAAT	CAGGTCAGCAGATTCACT
Solyc01g058010.2	*NI*	GCGTATAATCACTGGTAGC	GAATCCACTGCCTTCTTAG
Solyc03g078400.2	*Actin*	TGTCCCTATCTACGAGGGTTATGC	AGTTAAATCACGACCAGCAAGAT

**Table 3 foods-13-03599-t003:** Polyphenol composition was analyzed under different treatments.

Content (μg·g^−1^ DW)	CK0	CK	T1	T2
Quercetin	103.56 ± 2.65 c	462.50 ± 5.59 b	772.65 ± 6.18 a	612.32 ± 7.95 ab
Rutin	125.21 ± 8.87 ab	115.85 ± 9.64 b	156.69 ± 16.54 a	114.04 ± 2.82 b
Naringenin	2.31 ± 0.11 b	2.32 ± 0.20 b	3.57 ± 0.11 a	2.95 ± 0.31 ab
p-Hydroxybenzoic acid	9.22 ± 0.83 c	9.99 ± 0.57 c	17.74 ± 0.62 a	14.55 ± 1.24 b
Protocatechuic acid	8.39 ± 0.69 d	33.26 ± 3.01 c	76.79 ± 1.52 a	53.15 ± 1.74 b
Cinnamic acid	0.799 ± 0.08 b	1.16 ± 0.08 a	1.08 ± 0.04 a	1.02 ± 0.09 ab
4-coumalic acid	1.86 ± 0.08 c	2.60 ± 0.08 b	4.00 ± 0.16 a	2.10 ± 0.14 c
Gallic acid	50.379 ± 1.40 d	60.55 ± 0.73 c	93.22 ± 1.6 b	100.47 ± 3.5 a
Benzoic acid	123.93 ± 2.03 ab	126.06 ± 2.66 b	139.46 ± 1.52 a	113.42 ± 5.7 c
Ferulic acid	1.52 ± 0.25 c	1.32 ± 0.15 c	5.66 ± 0.13 a	2.32 ± 0.12 b
Caffeic acid	21.04 ± 0.96 ab	23.83 ± 0.92 a	20.30 ± 1.14 b	15.53 ± 0.62 c
Cynarin	26.54 ± 0.96 c	31.57 ± 1.32 bc	51.96 ± 4.08 a	37.35 ± 1.4 b
2,5-dihydroxybenzoic acid	155.51 ± 4.00 c	153.81 ± 3.16 c	554.16 ± 5.67 a	231.08 ± 5.18 b

Note: CK0 indicates no fertilizer, CK refers to conventional fertilizer based on local research, T1 denotes balanced fertilizer, and T2 represents a 10% reduction from the balanced fertilizer (10% less than T1). Values are the means of five biological replicates. Different letters indicate significant differences (Tukey’s test, *p* < 0.05).

**Table 4 foods-13-03599-t004:** The diverse volatile components present in tomato fruits.

Categories	Volatile Compounds	Chemical Formulas	Contents (µg·kg^−1^)			
			CK0	CK	T1	T2
Alcohols	Ethanol	C_2_H_6_O	293.95 ± 63.58 b	315.17 ± 30.18 b	496.97 ± 44.72 a	317.34 ± 13.51 b
	2-Octanol, (S)-	C_8_H_18_O	4.43 ± 0.69 b	4.77 ± 0.86 b	6.92 ± 0.33 a	2.8 ± 0.42 b
	1-Butanol, 3-methyl-	C_5_H_12_O	9.28 ± 0.55 b	7.44 ± 0.89 b	19.14 ± 0.42 a	15.45 ± 2.11 a
	Methane, isocyanato-	C_5_H_12_O	4.94 ± 0.94 b	6.24 ± 0.07 ab	8.62 ± 0.65 a	8.34 ± 1.13 a
	1-Penten-3-ol	C_5_H_10_O	1.73 ± 0.1 a	1.53 ± 0.05 a	2.58 ± 1.05 a	1.69 ± 0.14 a
	1-Pentanol	C_5_H_12_O	2.04 ± 0.87 c	7.24 ± 1.82 b	12.59 ± 0.66 a	2 ± 0.42 c
	1-Hexanol	C_6_H_14_O	106.57 ± 5.3 b	139.25 ± 44.49 b	326.23 ± 30.07 a	173.32 ± 26.88 b
	1-Ethynylcyclohexanol	C_8_H_12_O	1.3 ± 0.4 b	3.21 ± 0.68 a	3.49 ± 0.18 a	2.41 ± 0.09 ab
	(Z)-3-Hexen-1-ol	C_6_H_12_O	132.22 ± 6.88 b	145.45 ± 10.4 b	604.48 ± 48.38 a	515.16 ± 53.93 a
	Methyl nonyl ether	C_16_H_34_O	2.36 ± 0.5 b	2.23 ± 0.92 b	6.95 ± 0.27 a	4.17 ± 0.8 b
	1-Heptanol	C_7_H_16_O	n.d.	4.37 ± 0.21	4.02 ± 0.15	3.73 ± 0.11
	2-Octen-1-ol	C_8_H_16_O	2.13 ± 0.3 b	1.07 ± 0.08 b	6.34 ± 0.6 a	4.82 ± 0.88 a
	Hept-(4Z)-en-1-ol	C_7_H_14_O	0.33 ± 0.16 c	1.26 ± 0.14 bc	4.41 ± 0.4 a	2.58 ± 0.52 b
	Benzyl alcohol	C_7_H_8_O	0.54 ± 0.03 a	0.67 ± 0.28 a	0.97 ± 0.22 a	0.71 ± 0.16 a
	l-Phenylethyl Alcohol	C_8_H_10_O	1.03 ± 0.06 c	1.86 ± 0.46 bc	3.84 ± 0.39 a	2.56 ± 0.48 b
Aldehydes	2-Butenal, 2-methyl-, (E)-	C_5_H_8_O	3.3 ± 0.82 a	5.03 ± 0.44 a	6.06 ± 1.22 a	4.4 ± 1.19 a
	3-Hexenal	C_6_H_10_O	2.31 ± 0.54 b	4.32 ± 0.6 b	10.23 ± 1.06 a	4.08 ± 1.33 b
	Hexanal <n->	C_6_H_12_O	16.71 ± 1.74 b	22.43 ± 3.25 b	40.04 ± 4.73 a	41.78 ± 0.5 a
	(E)-2-Heptenal	C_7_H_12_O	n.d.	n.d.	2.23 ± 0.09	n.d.
	Pent-(2E)-enal	C_6_H_10_O	106.1 ± 10.28 b	131.09 ± 23.43 b	283.16 ± 15.84 a	232.62 ± 5.24 a
	Cyclohexane, 1,1’-(2-methyl-1,3-propanediyl) bis-	C_6_H_10_O	87.57 ± 3.69 b	133.50 ± 13.73 b	234.83 ± 7.5 a	225.09 ± 27.97 a
	2-Cyclohexen-1-ol	C_6_H_10_O	120.28 ± 10.28a	97.86 ± 23.43 b	89.1115.84 b	121.69 ± 5.24 a
	(E)-2-Hexenal	C_6_H_12_O	2.97 ± 0.58 b	1.89 ± 0.52 b	4.56 ± 0.4 a	1.81 ± 0.26 b
	Octanal	C_8_H_16_O	2.26 ± 1.01 a	2.11 ± 0.38 a	3.72 ± 1.86 a	1.33 ± 0.33 a
	2-Isobutylthiazole	C_6_H_10_O	1.94 ± 0.94d	2.30 ± 1.01c	3.43 ± 0.64a	2.87 ± 1.23b
	Nonanal	C_9_H_18_O	2.49 ± 0.51 b	2.14 ± 0.26 b	9.07 ± 0.39 a	3.3 ± 0.78 b
	Benzaldehyde	C_7_H_6_O	2.96 ± 0.64 b	3.56 ± 0.06 b	14.55 ± 2.34 a	6.22 ± 0.49 b
	(Z)-4-Decenal	C_10_H_18_O	n.d.	n.d.	1.37 ± 0.12	1.25 ± 0.09
Esters	Acetate <ethyl->	C_4_H_8_O_2_	3.69 ± 0.58 a	3.28 ± 0.5 a	4.11 ± 0.31 a	3.71 ± 0.52 a
	1-Butanol, 3-methyl-, acetate	C_7_H_14_O_2_	n.d.	3.21	3.21	2.75
	Hex-(3Z)-enyl acetate	C_8_H_14_O_2_	2.95 ± 1.1 b	3.4 ± 1.25 b	10.89 ± 0.38 a	3.84 ± 1.06 b
	Butyl salicylate	C_11_H_14_O_3_	1.58 ± 0.7 b	0.7 ± 0.47 b	4.64 ± 0.75 a	1.75 ± 0.47 b
	Methyl N-phenyl carbamate	C_8_H_9_NO_2_	n.d.	n.d.	40.72 ± 2.30	36.46 ± 1.29
Ketones	Penten-3-one	C_5_H_8_O	5.91 ± 0.51 b	6.45 ± 0.02 b	9.81 ± 0.88 a	6.92 ± 0.21 b
	Cycloheptadecanone	C_17_H_32_O	2.37 ± 0.55 b	2.75 ± 1.11 b	5.56 ± 0.55 a	1.47 ± 0.09 b
	2-Octanone	C_8_H_16_O	1.2 ± 0.24 b	2.88 ± 0.04 ab	5.13 ± 0.9 a	4.63 ± 1.67 ab
	Hept-5-en-2-one <6-methyl->	C_8_H_14_O	3.79 ± 2.03 c	13.03 ± 2.37 b	19.59 ± 0.77 a	12.11 ± 0.75 b
	2-Hydroxy acetophenone	C_8_H_8_O_2_	0.55 ± 0.19 a	0.31 ± 0.07 a	0.79 ± 0.39 a	0.4 ± 0.06 a
	Geranyl acetone	C_13_H_22_O	2.07 ± 0.93 b	5.38 ± 0.98 a	5.09 ± 0.61 ab	4.49 ± 0.91 ab
	trans-á-Ionone	C_13_H_20_O	0.33 ± 0.16 c	1.26 ± 0.14 bc	4.41 ± 0.4 a	2.58 ± 0.52 b
Hydrocarbons	p-Xylene	C_8_H_10_	n.d.	n.d.	n.d.	0.56 ± 0.01
	Tridecane	C_13_H_28_	n.d.	n.d.	n.d.	0.07 ± 0.01
Phenols	Phenol, 2-methoxy-	C_7_H_8_O_2_	2.64 ± 0.26 c	3.17 ± 0.19 c	11 ± 1.25 a	8.24 ± 1.09 b
	Eugenol	C_10_H_12_O_2_	0.62 ± 0.12 b	0.96 ± 0.21 ab	1.56 ± 0.12 a	1.27 ± 0.29 ab
	Carvacrol	C_10_H_14_O	0.6 ± 0.1 b	1.58 ± 0.41 ab	1.97 ± 0.12 a	1.14 ± 0.48 ab
Others	Furan <2-pentyl->	C_9_H_14_O	185.67 ± 17.93 b	186.47 ± 15.68 b	296.51 ± 42.66 a	249.9 ± 20.22 ab
	2-Isobutylthiazole	C_7_H_11_NS	4.13 ± 0.85 b	3.46 ± 0.56 b	9.12 ± 0.39 a	1.72 ± 0.85 b
Total content (µg·kg^−1^)			1129.84	1286.28	2644.02	2047.53

Notes: CK0 indicates no fertilizer, CK refers to conventional fertilizer based on local research, T1 denotes balanced fertilizer, and T2 signifies a 10% reduction from the balanced fertilizer (10% less than T1). Values represent the mean of five biological replicates, with different letters indicating significant differences (Tukey’s test, *p* < 0.05). “n.d.” indicates no data, meaning the compounds were not detected. Values are the means of five biological replicates. Different letters indicate significant differences (Tukey’s test, *p* < 0.05).

**Table 5 foods-13-03599-t005:** Changes in characteristic volatile compounds’ contents in tomato fruit.

Characteristic Aroma Component	Aroma Type	Content (µg·kg^−1^)			
		CK0	CK	T1	T2
1-Penten-3-ol	Fruity	1.73 ± 0.1 a	1.53 ± 0.05 a	2.58 ± 1.05 a	1.69 ± 0.14 a
(Z)-3-Hexen-1-ol,	Grassy, green	132.22 ± 6.88 b	145.45 ± 10.4 b	604.48 ± 48.38 a	515.16 ± 53.93 a
l-Phenylethyl Alcohol	Floral	1.03 ± 0.06 c	1.86 ± 0.46 bc	3.84 ± 0.39 a	2.56 ± 0.48 b
3-Hexenal	Grassy, green	2.31 ± 0.54 b	4.32 ± 0.6 b	10.23 ± 1.06 a	4.08 ± 1.33 b
Hexanal <n->	Grassy, green	16.71 ± 1.74 b	22.43 ± 3.25 b	40.04 ± 4.73 a	41.78 ± 0.5 a
(E)-2-Heptenal	Grassy	n.d.	n.d.	2.23 ± 0.09	n.d.
(E)-2-Hexenal	Grassy, green	2.97 ± 0.58 b	1.89 ± 0.52 b	4.56 ± 0.4 a	1.81 ± 0.26 b
Penten-3-one	Fruity	5.91 ± 0.51 b	6.45 ± 0.02 b	9.81 ± 0.88 a	6.92 ± 0.21 b
Hept-5-en-2-one <6-methyl->	Fruity	3.79 ± 2.03 c	13.03 ± 2.37 b	19.59 ± 0.77 a	12.11 ± 0.75 b
trans-á-Ionone	Fruity, aromatic	0.33 ± 0.16 c	1.26 ± 0.14 bc	4.41 ± 0.4 a	2.58 ± 0.52 b
2-Isobutylthiazole	Fruity, green	4.13 ± 0.85 b	3.46 ± 0.56 b	9.12 ± 0.39 a	1.72 ± 0.85 b

Different letters indicate significant differences (Tukey’s test, *p* < 0.05).

## Data Availability

The original contributions presented in the study are included in the article, further inquiries can be directed to the corresponding author.

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
