# Peer review of "Balanced Fertilization Enhances the Nutritional Value and Flavor Profile of Tomato Fruits"

_foods, 2024, doi:10.3390/foods13223599_

Round 1
Reviewer 1 Report
Comments and Suggestions for Authors
Dear author,
Thanks for your good report. My questions and comments are in the attached file.
Regards

Reviewer 2 Report
Comments and Suggestions for Authors
The article is, in our opinion, highly original because of the approach concerning the flavor profile of tomato fruits.
Abstract
Please explain the abbreviation CK.
Introduction
The introduction is well documented, but low space is allowed to the discussion of tomato flavor. Thus, we recommend authors to bring updates concerning this issue. We also recommend authors to mention in a clear manner the aim of their study. As mentioned, it does not cover all issues discussed in the article (i.e. the reason why they perform qRT-PCR analysis).
Materials and Methods
In this section the materials and methods are well described and explained.
Results
The results are clearly presented.
Discussions
The discussions are explanatory for all addressed issues.
Conclusions
The conclusions are in concordance with addressed issues.
Round 2
Reviewer 1 Report
Comments and Suggestions for Authors
Dear author,
Thanks for your revised paper. Regarding my question as follow, please add this reference to your paper and explain your previous findings to your text too.
Regards
We need plant and fruit analysis for minerals (main macro and micro elements) to figure it out clearly which elements influence plant growth and fruit quality. So, as there is no fruit elements analysis the discussion section is not strong.
Response: Thank Reviewer so much for the considerable reminding! Regarding the determination of mineral elements in tomato fruits under balanced fertilization treatment, our research group has already conducted relevant studies in related papers and clearly reached the conclusion that balanced fertilization increased the contents of major and minor mineral elements in plants and fruits. The references are as following:
[1] Li Wangxiong,Zhang Yang,Tang Zhongqi,Yu Jihua. Effects of balanced fertilization on growth,quality,mineral elements contents and yield of tomato cultivated in substrate in greenhouse [J]. Acta Agriculturae Zhejiangensis, 2022, 34 (08): 1648-1660.
